# VISTA Is a Diagnostic Biomarker and Immunotherapy Target of Aggressive Feline Mammary Carcinoma Subtypes

**DOI:** 10.3390/cancers13215559

**Published:** 2021-11-05

**Authors:** Andreia Gameiro, Catarina Nascimento, Jorge Correia, Fernando Ferreira

**Affiliations:** CIISA—Centro de Investigação Interdisciplinar em Sanidade Animal, Faculdade de Medicina Veterinária, Universidade de Lisboa, Avenida da Universidade Técnica, 1300-477 Lisboa, Portugal; agameiro@fmv.ulisboa.pt (A.G.); catnasc@fmv.ulisboa.pt (C.N.); jcorreia@fmv.ulisboa.pt (J.C.)

**Keywords:** feline mammary carcinoma, breast cancer, VISTA, immune checkpoint, cancer model

## Abstract

**Simple Summary:**

Mammary tumors are common in cats, showing aggressive behavior and few therapeutic options. Recently, feline mammary carcinomas have become a reliable cancer model for human breast cancer studies, due to the similarities between the two species. Thus, the identification of new tumor biomarkers and therapeutic targets to improve cat’s prognosis is needed. VISTA is an important immune checkpoint protein that has gained importance over the past few years in women’s cancers. In this study, the serum VISTA levels and tumor expression were analyzed in cats with mammary tumors, being correlated with other immune checkpoints. In the diseased animals, VISTA is overexpressed in more aggressive tumor subtypes (HER2-positive and triple-negative), showing a positive correlation with the expression of VISTA in tumor-infiltrating lymphocytes, and is associated with an immunosuppressive status, suggesting that VISTA could be a promising non-invasive prognostic biomarker and therapeutic target in cats with mammary carcinomas, as reported in humans.

**Abstract:**

Feline mammary carcinoma (FMC) is a common neoplasia, showing aggressive clinicopathological features, without viable therapeutic options. The study of tumor microenvironment has gained importance, due to the ability to control tumor progression by regulating the immune response. Considering the lack of knowledge, feline serum VISTA levels from cats with mammary carcinoma were compared with healthy controls, and with serum levels of PD-1/PD-L1, CTLA-4, LAG-3, IL-6, and TNF-α. In parallel, VISTA tumor expression was evaluated in FMC samples. The obtained data revealed that serum VISTA levels were significantly higher in cats presenting HER2-positive (*p =* 0.0025) or triple-negative subtypes (*p =* 0.0019), with higher serum levels in luminal A (*p =* 0.0025) correlated to the presence of metastasis (*p =* 0.0471). Furthermore, in HER2-positive or triple-negative tumors, correlations were obtained between serum VISTA levels and the serum levels of the above-mentioned molecules. In tumors, VISTA expression revealed a stronger intensity in cancer cells, when compared to TILs (*p <* 0.0001). Stratifying the samples by subtypes, a higher number of VISTA-positive TILs was observed in the HER2-positive subtype, compared with triple-negative tumors (*p =* 0.0138). In conclusion, results support the development of therapeutic strategies for HER2-positive and triple-negative FMC subtypes, reinforcing the use of cats as a human oncology model.

## 1. Introduction

Feline mammary carcinoma (FMC) is the third most common tumor in cats, representing 12% to 40% of all neoplasms [1,2] with an aggressive and infiltrative behavior [1,2,3], as in human breast cancer [4,5]. At the molecular level, human and feline breast cancer could be categorized into different subtypes, listed by its malignant behavior: luminal A, luminal B, HER2-positive, and triple-negative [6,7]. Briefly, considering the most aggressive subtypes, the HER2-positive subtype is defined by the overexpression of HER2 and lack of hormonal receptors, such as estrogen (ER) and/or progesterone (PR), presenting an incidence of 33% to 60% [2,8]. In parallel, the triple-negative subtype is characterized by the absence of HER2, ER, and PR, representing 37% to 54% [9,10,11] of all FMC.

In cats, a lack of therapeutic options, and in some cases an ineffective chemotherapy protocol (vincristine, cyclophosphamide, doxorubicin) [12] makes the mastectomy crucial for the FMC treatment, being urgent the development of different therapeutic options in order to improve the clinical outcome of these animals. Moreover, in humans, immunotherapy has emerged as a promising tool in the treatment of certain types of breast cancer by using the self-immune system to attack cancer cells against tumor-specific antigens [13,14]. This strategy can use several immune checkpoint molecules involved in inhibitory pathways, modulating the immune responses [13,15]. Thus, cancer immunotherapy targeting proteins including programmed cell death-1 (PD-1) and programmed cell death ligand-1 (PD-L1), lymphocyte activation gene 3 (LAG-3), and cytotoxic T-lymphocyte-associated antigen-4 (CTLA-4), proved to be effective in different types of cancers [16,17,18]. In fact, the PD-1/PD-L1 axis plays an important role in tumor immune response, in both human and feline breast cancers [19,20], and is up-regulated together with LAG-3 in human breast cancer [21]. Furthermore, the CTLA-4 is an immune mediator that inhibits T-cell immune function, being increased in humans and cats with malignant mammary tumors [22]. In addition, V-domain immunoglobulin suppressor of T cell activation (VISTA) arises as a novel immune checkpoint regulator [13,23]. Accordingly, VISTA is able to inhibit T cell activation, maintaining tolerance and limiting immunopathology [24], acting as a ligand on antigen-presenting cells, and as a receptor on T cells [25,26]. In parallel, in human tumor samples, the VISTA protein is described as being highly expressed on tumor-infiltrating lymphocytes (TILs) and myeloid cells [27,28], and is associated with a good prognosis [28,29]. Furthermore, the evaluation of VISTA expression in tumor samples may be a potential biomarker used to decide therapy targets [13]. In fact, in murine cancer models, the inhibition of VISTA weakens the suppressive function of T cells, resulting in tumor growth reduction [13,23,24,30]. This observation suggests the VISTA protein has a potential target for cancer therapy [13,29,30,31], being tested in a phase I trial to treat patients with solid tumors, by targeting VISTA and PD-L1/PD-L2 proteins (NCT02812875) [32]. Moreover, recent clinical trials have revealed its value for cancer treatment by using monoclonal antibodies and small molecules targeting VISTA (Janssen Inc., NCT02671955; Curis Inc., NCT02812875) [33].

In order to surpass the limitations of laboratory rodents, the use of domestic animal models with spontaneous tumors has increased, allowing researchers to understand the tumorigenesis process [34,35]. Despite cats being considered a valuable human model for different diseases [3,36,37], efforts are needed for a deeper understanding of the FMC development, not only to improve cats’ survival time but also to reinforce its use as an oncology model. Thus, to contribute to knowledge regarding VISTA expression in FMC progression and immune response, this study aimed to: (i) quantify and compare serum VISTA levels in cats with different subtypes of mammary tumors and healthy controls; (ii) test for statistical associations between serum VISTA levels and the previously reported serum PD1-/PD-L1, CTLA-4, LAG-3, interleukin 6 (IL-6) and tumor necrosis factor-α (TNF-α) levels; (iii) evaluate VISTA expression in TILs and cancer cells; and (iv) correlate serum and tissue VISTA levels with animals’ clinicopathological features, in order to understand the utility of VISTA as diagnostic and/or prognostic biomarker or novel therapeutic target for cats with mammary carcinoma.

## 2. Materials and Methods

### 2.1. Animal Population

After a fully recorded clinical history and FMC’s diagnosis, tumor tissue and serum samples were collected from 46 queens. Briefly, the animal population exhibited a mean age of 11.9 years (range 6.5–18 years) at the time of diagnosis. The disease-free survival (DFS) and the overall survival (OS) rates were also recorded, being 8.9 ± 7.5 months (*n* = 45; 95% CI: 6.5–11.2 months) and 14.3 ± 9.6 months (*n* = 42; 95% CI: 11.3–17.3 months), respectively. As controls, 14 serum samples were collected from healthy cats that underwent elective ovariohysterectomy at the Teaching Hospital of the Faculty of Veterinary Medicine, University of Lisbon, Portugal, presenting a mean age of 1 year (range 0.5–1.5 years). The animals were anesthetized before surgery and blood samples collection was performed without interfering with their wellbeing, with all the manipulation procedures being consented to by the owners.

In addition, the clinicopathological data for all the animals were registered. Briefly, the following features were recorded: breed; age; reproductive status; applied treatment; presence of multiple tumors, its size, and histopathological classification; hormonal (ER and PR) and HER2 status [6] and Ki-67 index [38] of the tumor masses; malignancy grade, scored based on the Elston & Ellis system [39]; presence of tumor necrosis, positive lymphatic invasion or lymphocytic infiltration, and/or cutaneous ulceration; involvement of the regional lymph node; and clinical stage (TNM system; Table 1). Moreover, the molecular subtyping of FMC [2,6] were characterized, with cats being stratified into four different groups: luminal A (*n* = 7), luminal B (*n* = 16), HER2-positive (*n* = 10) and triple-negative (*n* = 13) subtypes.

The collected tumor tissue samples were subjected to a controlled fixation process for 24 to 48 hours using 10% buffered neutralized formalin (pH 7.2) and embedded in paraffin. In parallel, blood samples were centrifuged (1,500× *g*, 10 min, 4 °C), in order to separate the serum from clotted blood, and stored at −80 °C until further use. As recommended, after thawed, all samples were observed and discarded, when presented hemolysis [2,40].

Additionally, the homology between the immunogen location in human and feline VISTA proteins is 93.0% (UniProt, accession numbers: *Homo sapiens* Q9H7M9; *Felis catus* M3WI46).

### 2.2. Measurement of Serum VISTA Levels

To quantify the serum VISTA levels a commercial ELISA-based kit (R&D Systems, Minneapolis, USA; DY7126-05) was used and the manufacturer’s instructions were followed. Firstly, using the recombinant proteins provided in the kit, a standard curve was plotted with 12-fold serial dilutions, and by a quadratic regression, the *r*^2^ value was calculated (*r*^2^ = 0.9959). Summarizing the technique, a 96-well plate was coated with the capture antibody and incubated overnight. After this initial step, and to prevent non-specific protein binding, plates were incubated for 1 hour with 1% bovine serum albumin (BSA) in phosphate-buffered saline (PBS). Then, standards and diluted serum samples of controls and diseased animals were added to the plate and incubated for 2 hours, followed by another 2-hour incubation step with the detection antibody, both at room temperature (RT). To conclude the procedure, the streptavidin conjugated to horseradish peroxidase (HRP) was added to the plate and incubated for 20 min at RT, and then the substrate solution in 1:1 H_2_O_2_ and tetramethyl-benzidine (20 min at RT in the dark). To interrupt the reaction, a stop solution (2NH_2_SO_4_) was mixed in each well and the absorbance was read by a spectrophotometer (FLUOStar OPTIMA, Microplate Reader, BMG, Ortenberg, Germany), using a 570 nm reference wavelength and a primary wavelength of 450 nm.

Furthermore, to conclude about the systemic response of the diseased queens, data on immune checkpoint molecules and cytokines (PD-1/PD-L1, CTLA-4, LAG-3, IL-6 and TNF-α), were considered in this study. The measurement of this molecules presented a similar procedure, as previously reported [19,22].

### 2.3. Assessment of VISTA Expression by Immunohistochemistry (IHC)

Initially, to define a representative tumor area, the formalin-fixed paraffin-embedded (FFPE) tumor tissue samples (*n* = 46) were stained with hematoxylin–eosin. Then, 3 μm slices of the selected FFPE tumor areas were sectioned (Microtome Leica RM135, Newcastle, UK) and mounted on a superfrost glass slide (SuperFrost Plus, Thermo Fisher Scientific, MA, USA). Samples were deparaffinized, hydrated, and antigen retrieval, using citrate buffer pH 6.0 (EnVisionTM Flex Target Retrieval Solution High pH, DAKO) was performed on a PT-Link module (DAKO, Agilent, Santa Clara, CA, USA) for 20 min at 96 °C. Thereafter, slides were cooled until reaching RT and immersed twice for 5 min in distilled water. For the IHC technique solutions from the commercial kit, the Novolink TM Max Polymer Detection System Kit (Leica Biosystems, Newcastle, UK) was used. Firstly, samples were incubated for 15 min with Peroxidase Block Novocastra Solution (Leica Biosystems), followed by blocking the unspecific antigenic recognition for 10 min using Protein Block Novocastra Solution (Leica Biosystems). Then, the anti-VISTA primary polyclonal antibody (Ref.: ab254581; Abcam, Cambridge, UK) diluted at 1:200 was incubated was added in a humidified chamber overnight at 4 °C, as recommended [28,29]. After this period, the detection polymer was incubated at RT for 30 min, followed by the detection solution of diaminobenzidine (DAB substrate buffer and DAB Chromogen, Leica Biosystems) for 5 min. During this technique, between all the incubation steps, the slides were washed twice for 5 min, using a PBS solution pH 7.4. To finish the IHC, samples were counterstained for 2 min with Gills hematoxylin (Merck, NJ, USA), dehydrated in an ethanol gradient and xylene, and mounted using Entellan mounting medium (Merck). As positive controls, to validate the technique, human tonsil and feline lymph nodes were used, whereas feline lymph nodes were also used as internal controls. In parallel, healthy tissue sections in the analyzed samples were evaluated as negative controls.

VISTA expression was scored according to the guidelines described by the International TILs Working Group 2014 applied to breast cancer. In sum, tumor tissue sections were evaluated, considering the area occupied by lymphocytes [41] in the stromal compartment, and excluding zones with necrosis and the TILs outside the tumor areas. Additionally, VISTA immunostaining was evaluated in membranes and the nuclei of the TILs, which were identified by their characteristic morphology. Furthermore, cancer cells were evaluated for VISTA immunostaining. The percentage of stained cells was defined as: 0% meaning the absence of staining to 100% implying that all cells were stained [29,42], with a cut-off value established at 5% [43] for a positive sample. Additionally, VISTA intensity was evaluated in the tumor samples to assess the immunostaining distribution. As previously reported [28,44,45], the staining intensity was classified by the following scores: 0 (no signal), 1+ (weak), 2+ (moderate) and 3+ (intense). The IHC slides were classified by two pathologists, in a blind scoring evaluation.

### 2.4. Data Statistical Analysis

The statistical analysis of the data collected in this study was performed using the GraphPad Prism software, version 5.04 (California, USA). Statistically significant results were considered when two-tailed *p*-values < 0.05 were observed, considering a 95% confidence interval (* *p* < 0.05, ** *p* < 0.01 and *** *p* < 0.001). The calculated mean values were reported with the proper standard error (SEM). The non-parametric Mann–Whitney test was used to compare the serum VISTA levels with several clinicopathological features. In order to choose the optimal cut-off values for serum VISTA levels, and to determine the specificity and sensitivity of the technique to properly identify tumor-positive animals, the receiver-operating characteristic (ROC) curves were calculated. Furthermore, using the Kaplan–Meier test, survival analysis was performed to evaluate the DFS in diseased queens. In parallel, to infer cats’ immunosuppressive status, correlations between the previously reported serum PD-1/PD-L1 [19], CTLA-4, IL-6, TNF-α [22], and LAG-3 levels and the serum VISTA concentrations determined in this study were investigated applying the Spearman’s rank correlation coefficient. To end the analysis and compare VISTA serum levels and tissue expression status in cats with mammary carcinoma, the non-parametric Kruskal–Wallis test was applied.

## 3. Results

### 3.1. Serum VISTA Levels Are Higher in Cats with Luminal A, HER2-Positive or Triple-Negative Tumor Subtypes

Serum VISTA levels were evaluated in a healthy/control group and compared to those of a population of queens diagnosed with mammary carcinoma, stratified by its molecular subtypes (Table 2).

The statistical analysis revealed differences between the mean ranks of at least one pair of groups (*p =* 0.0016; Figure 1A). In fact, cats showing luminal A, HER2-positive or triple-negative tumor subtypes presented higher serum VISTA levels than the healthy population (10.31 ± 1.09 pg/mL. vs. 5.90 ± 0.51 pg/mL, *p =* 0.0025; 12.90 ± 3.73 pg/mL vs. 5.90 ± 0.51 pg/mL, *p =* 0.0025; 14.36 ± 4.16 pg/mL vs. 5.90 ± 0.51 pg/mL, *p =* 0.0019, respectively).

Additionally, the optimal cut-off value of serum VISTA levels was 7.348 pg/mL in cats with the less aggressive tumor subtype (luminal A; *p =* 0.0022, sensitivity = 100% and specificity = 85.71%; AUC = 0.9184 ± 0.0623, 95% CI: 0.7963–1.040; Figure 1B), and for the most aggressive tumor subtypes, cats with HER2-positive carcinoma presented a cut-off value of 7.231 pg/mL (*p =* 0.0023, sensitivity = 80.00% and specificity = 85.71%; AUC = 0.8714 ± 0.0727, 95% CI: 0.7290–1.014; Figure 1C) and 6.765 pg/mL for cats presenting triple-negative tumors (*p =* 0.0018, sensitivity = 84.62% and specificity = 78.57%; AUC = 0.8544 ± 0.0768, 95% CI: 0.7038–1.005; Figure 1D).

Furthermore, the correlation of serum VISTA levels with the recorded clinicopathological features, revealed that for the luminal A tumor subtype, an increase in serum VISTA levels was associated with the metastization process (*p =* 0.0471; Figure 2). 

### 3.2. Cats with Mammary Tumors, Namely HER2-Positive or Triple-Negative Subtypes, Presented a Positive Correlation between Serum VISTA Levels and Serum Levels of Distinct Immune Checkpoint Molecules and Cytokines

Considering the tumor population, several correlations were observed between serum VISTA levels and distinct immune checkpoint molecules, such as serum PD-1 (*p <* 0.0001, *r* = 0.6523; Figure 3A) and PD-L1 (*p <* 0.0001, *r* = 0.7434; Figure 3B) levels [19], serum CTLA-4 (*p <* 0.0001, *r*= 0.6136; Figure 3C) levels [22] and serum LAG-3 *p =* 0.0002, *r* = 0.5336; Figure 3D) levels. Additionally, a positive correlation between serum VISTA levels with serum IL-6 (*p =* 0.0016, 0.4525; Figure 3E) and TNF-α (*p <* 0.0001, *r* = 0.6220; Figure 3F) levels was also found [22].

Specifically, animals with HER2-positive tumor subtype revealed a positive correlation between serum VISTA levels and the previously reported serum PD-1 levels [19] (Table 3). Additionally, a correlation was also observed between serum VISTA levels and PD-1 and PD-L1 [19], CTLA-4 [22], LAG-3, IL-6, and TNF-α levels [22], in cats presenting triple-negative tumors.

### 3.3. The Tumor VISTA Analysis Revealed a Heterogeneous Intensity along the Cancer Cells, While TILs Presented a Homogeneous Intensity in the General of the Tumor Samples

Tissue VISTA expression was evaluated in TILs and cancer cells of FMC samples, stratified by different subtypes (Table 4).

In order to evaluate feline VISTA expression, human tonsil and feline lymph nodes were used as positive controls, revealing a similar staining pattern, with a strong IHC score (3+) in lymphocytes (Figure 4 A,B, respectively). Furthermore, normal mammary cells, as well as the tumor interstitial compartment, namely fibroblasts and collagen cells revealed no VISTA expression (0 score; Figure 4 C,D, respectively). The tumor tissue areas showed a heterogeneous intensity in the VISTA staining, in the general of the cancer cells, contrary to a homogenous VISTA intensity in TILs, independent of the tumor subtype (Figure 4E,F).

In addition, FMC samples presented a different VISTA expression pattern along the tumor. A stronger VISTA intensity was observed in the epithelial luminal zones (3+ score) of the cancer cells (Figure 5), when compared to the mesenchymal areas. Furthermore, a stronger VISTA intensity was observed in the membrane, when compared with the cell cytoplasm.

Moreover, the results revealed that VISTA-positive cancer cells and serum VISTA levels were found in 100% of tumors, independent of the molecular subtype. In parallel, VISTA-positive TILs and serum VISTA levels occurred in 82.61% (38/46 samples) of the studied population. Thus, by analysis of the tumor tissue samples, evaluating VISTA expression in TILs and cancer cells, and comparing it to the serum VISTA levels measured, differential and complementary information could be obtained.

### 3.4. VISTA Expression Is Higher in Cancer Cells, with VISTA-Positive TILs Being Associated with Less Aggressive Clinicopathological Features

A higher percentage of VISTA-positive cancer cells were observed in tumor areas, when compared to TILs (*p <* 0.0001; Figure 6A). Interestingly, when VISTA intensity is multiplied by the percentage of positive cells, a correlation was observed in the VISTA expression between cancer cells and TILs (*p =* 0.0022, *r* = 0.4543; Figure 6B). Moreover, the percentage of VISTA-positive TILs occurred higher in malignancy grade II (*p =* 0.0025; Figure 6C), with no TILs present in malignancy grade I.

### 3.5. VISTA Expression Is Higher in HER2-Positive Tumors Than in Triple-Negative Carcinomas

Stratifying tumors by their subtype, a significant difference of VISTA-positive TILs expression was observed between HER2-positive and triple-negative tumors (*p =* 0.0138; Figure 7A), with no differences in VISTA-positive cancer cells (*p =* 0.4677; Figure 7B).

Furthermore, considering the HER2-positive tumor subtype and evaluating the cancer cells, a higher VISTA expression was obtained in tumors with a metastization process (*p =* 0.0200; Figure 8).

## 4. Discussion

The majority of feline mammary tumors are malignant, with the HER2-positive and triple-negative subtypes presenting a poor prognosis [6,46]. Therefore, early detection and effective treatments will have a major impact on survival time [47], with the development of more efficient therapies being imperative. Furthermore, the immune checkpoint proteins produced by cancer cells, as a strategy to escape from the immune system, downregulating the systemic response, could reveal as important therapeutic targets. Currently, there are several proteins from the immune system, targeted as adjuvants in the treatment of human breast cancer, with some of these already proposed to be tested in diseased cats [19,22]. However, resistance mechanisms to immune checkpoint inhibitors have already been described, e.g., to anti-PD-L1 and anti-CTLA-4 compounds [48,49], being important to have different therapeutic options for the improvement in cats’ prognosis. Thus, a better understanding of the VISTA expression in diseased animals and FMC samples could reveal not only new non-invasive diagnostic or prognostic biomarkers but also improved options as therapeutic targets for cats with mammary carcinoma. 

For the first time, serum VISTA levels were evaluated in a diseased population, and queens diagnosed with the most aggressive mammary carcinoma subtypes, HER2-positive or triple-negative [6], revealed higher serum VISTA expression levels (*p =* 0.0025 and *p =* 0.0019, respectively). These results are in concordance with the findings described for human breast cancer [21,29,50], suggesting a conserved role of the VISTA protein between both species. Interestingly, in cats with luminal A tumor subtype, an increase in serum VISTA levels (*p =* 0.0025) was also observed, being associated with the metastization process (*p =* 0.0471). This result suggests the need for the VISTA protein for the escape and evasion of cancer cells from the immune system [44,51,52,53].

The VISTA protein is known as being responsible for the resistance to anti-PD-1/anti-CTLA-4 treatments, reported in prostate cancer [54,55], with an overall success rate of 30% of these targeted therapies [17,28,56], suggesting that blocking VISTA could be a good strategy to improve patients’ prognosis. Thus, and as reported in the tumor microenvironment (TME) of distinct human cancers [21,28,29,50,57], positive correlations were observed between serum VISTA levels and several immune inhibitory checkpoint proteins [58], such as serum PD-1 (*p <* 0.0001), PD-L1 (*p <* 0.0001), CTLA-4 (*p <* 0.0001), and LAG-3 (*p =* 0.0002), as well as with the cytokines IL-6 (*p =* 0.0016) and TNF-α (*p <* 0.0001), proposing not only a co-regulation among these molecules [20,55,59,60] but also an immunosuppressive status of these animals, reinforcing previously reported data [19,22]. Furthermore, cats with triple-negative carcinomas presented a positive correlation with all the immune checkpoint proteins and cytokines described, suggesting that this tumor is highly immunogenic, as in humans [43].

In human cancers, VISTA plays a crucial role in the anti-tumor immunity and potentiates a local immunosuppressive status [28,30], its overexpression occurring in the TME [30], presenting immunostaining in the membrane and/or cell cytoplasm [43]. In parallel, in feline tumor samples, TILs present a VISTA-positive expression in the cell membrane and cytoplasm, while in VISTA-positive cancer cells, the protein’s expression occurs mainly in the cell membrane, when compared to the cell cytoplasm. The cytoplasmic staining may represent intracellular stores of VISTA that will be released with the appropriate stimulation [44,61]. Furthermore, VISTA had a specific spatial/geographical pattern [25], similar to what was obtained in the feline tumors. In these samples, a homogenous VISTA expression was observed in TILs, while in cancer cells a heterogeneous VISTA expression was obtained, according to the findings in human cancers. Thus, this result indicates different cancer cell subpopulations that vary in their genetic, phenotypic, or behavioral characteristics, being associated with cancer progression and therapeutic resistance [62,63].

The statistical analysis of the immunostaining in the FFPE FMC samples revealed a higher expression of VISTA in cancer cells when compared to TILs (*p <* 0.0001), contrary to what is reported in a variety of human cancers [29,32,43,44,63,64]. Interestingly, a positive correlation was obtained by the increase in VISTA expression in cancer cells and TILs (*p =* 0.0022), which suggests a paracrine mechanism used by cancer cells to communicate with the immune system, as reported for PD-L1 in breast cancer patients [65,66,67]. Furthermore, a higher expression of VISTA-positive TILs occurs in grade II FMC, compared to grade III (*p =* 0.0025), which is in accordance with findings in human breast cancer, with VISTA TILs’ expression being associated with a good prognosis [28,29]. Furthermore, TILs-positive VISTA expression was higher in HER2-positive carcinomas, contrasting with triple-negative subtype (*p =* 0.0138), as reported in human breast cancer [21,28,29,50]. In addition, previous studies in breast cancer patients revealed that VISTA leads to a local immunosuppressive microenvironment [44,51,52,53], strengthening the results reported in this study, and the systemic immunosuppressive status described in queens diagnosed with HER2-positive tumors [6,19,22]. Interestingly, in the HER2-positive subtype, VISTA-positive cancer cells were associated with the tumor metastization process (*p =* 0.0200), suggesting that cancer cells release VISTA in order to downregulate the tumor immune response [44,51,52,53] and to evade the primary tumor location. 

Notably, despite the lower concentration of serum VISTA levels found in the diseased animals, when compared to other proteins reported [19,22], an optimal cut-off value for VISTA was possible to determine, presenting higher sensibility, when compared, e.g., to PD-1 and PD-L1 [19]. Furthermore, the high concordance rate found between serum VISTA levels and VISTA-positive cancer cells in the FMC cases suggests VISTA protein as a non-invasive diagnostic biomarker, as reported for breast cancer patients [42,68].

## 5. Conclusions

In conclusion, these results demonstrated that higher serum VISTA levels were correlated with the most aggressive tumor subtypes (HER2-positive or triple-negative) and poor clinicopathological features, as the association to the metastization process in luminal A subtype. Furthermore, in cats presenting HER2-positive mammary carcinomas, an association was observed comparing the VISTA and PD-1 levels in circulation. In parallel, queens with triple-negative mammary tumors demonstrated correlations between different immune checkpoint proteins, such as PD-1/PD-L1, CTLA-4, and LAG-3, as well as between the cytokines IL-6 and TNF-α, suggesting a systemic immunosuppression of these cats. Additionally, the microenvironment of HER2-positive tumors revealed a high VISTA expression in TILs when compared to the triple-negative subtype, being the VISTA-cancer cells expression associated with the metastization process. Altogether, the results obtained suggest that diseased cats presenting the most aggressive tumor subtypes could take advantage of resorting to immune-targeted therapies (e.g., anti-PD-1, anti-CTLA-4, or anti-VISTA). Moreover, the similarities unveiled between cat and human breast cancers provide rational support to consider the spontaneous FMC as a valuable model in oncology studies.

## Figures and Tables

**Figure 1 cancers-13-05559-f001:**
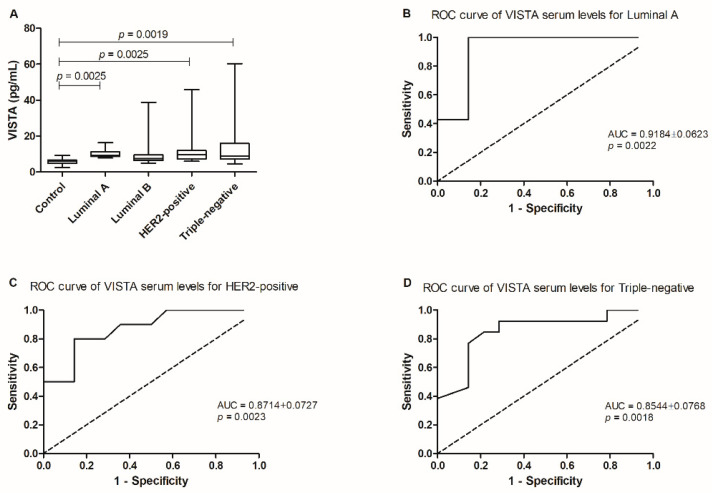
Serum VISTA levels in the diseased population in the study by comparison to a control/healthy group. (**A**) Serum VISTA levels were higher in queens with luminal A (* *p =* 0.0025), HER2-positive (* *p =* 0.0025) and triple-negative (** *p =* 0.0019) tumor subtypes, when compared to the control group. (**B**) ROC curve analysis performed in tumor subtypes presenting the higher serum VISTA levels allowed the identification of an optimal cut-off value for cats diagnosed with luminal A; and the most aggressive, (**C**) HER2-positive; (**D**) and triple-negative carcinomas.

**Figure 2 cancers-13-05559-f002:**
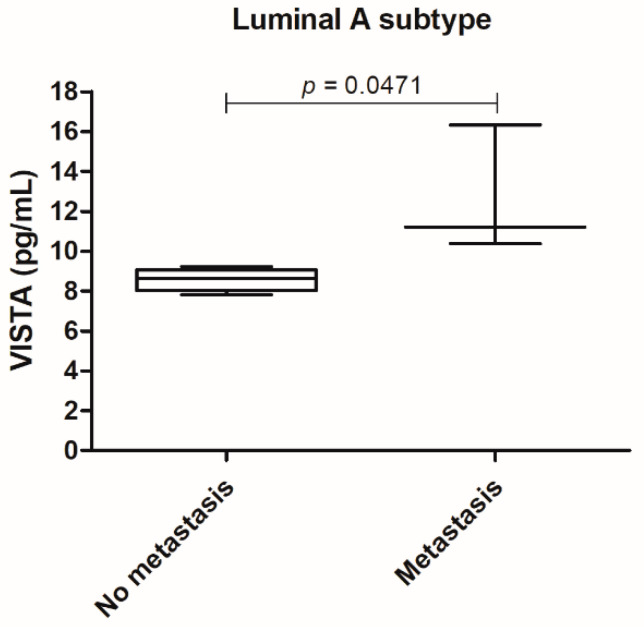
In the luminal A tumor subtype serum VISTA levels were higher in cats presenting metastasis (* *p =* 0.0471).

**Figure 3 cancers-13-05559-f003:**
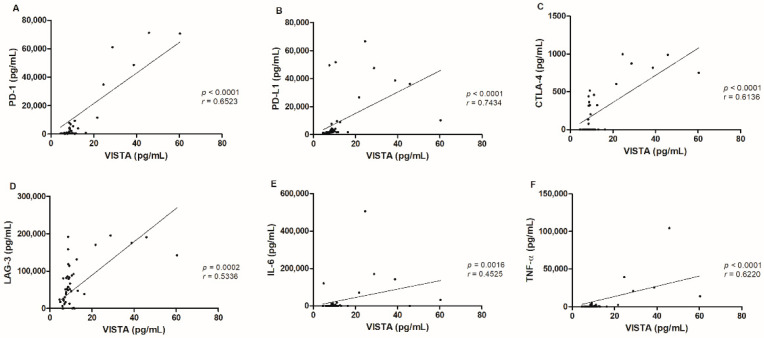
Serum VISTA levels in cats with mammary carcinoma were correlated with several immune checkpoint proteins, namely (**A**) PD-1 (*** *p <* 0.0001), (**B**) PD-L1 (*** *p <* 0.0001), (**C**) CTLA-4 (*** *p <* 0.0001), (**D**) LAG-3 (*** *p =* 0.0002), (**E**) IL-6 (***p =* 0.0016) and (**F**) TNF-α (****p <* 0.0001) serum levels.

**Figure 4 cancers-13-05559-f004:**
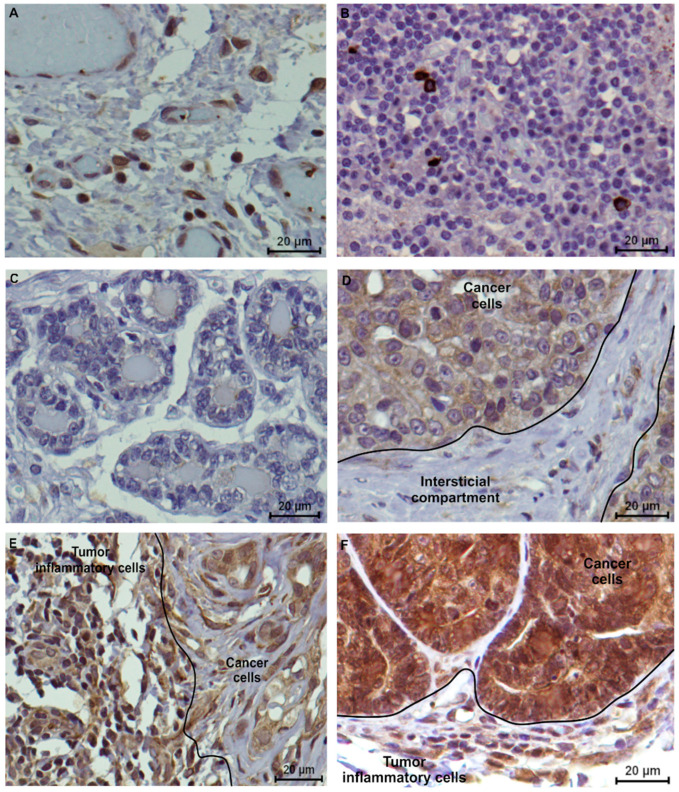
VISTA immunostaining in controls and feline tumor samples (400x magnification). (**A**) Human tonsil and (**B**) feline lymph nodes were used as positive controls, presenting a strong (score 3+) tissue VISTA expression in lymphocytes; (**C**) Considering the feline tumor samples, normal mammary cells and (**D**) tumor interstitial compartment revealed a negative VISTA expression. (**E**) Triple-negative (evaluated as 70% of positive TILs, with 2+ score and 60% of positive cancer cells, with 2+ score) and (**F**) HER2-positive tumor subtypes (evaluated as 80% of positive TILs, with 2+ score and 100% of positive cancer cells, with 3+ score) were used to demonstrate the VISTA tissue expression in feline tumors.

**Figure 5 cancers-13-05559-f005:**
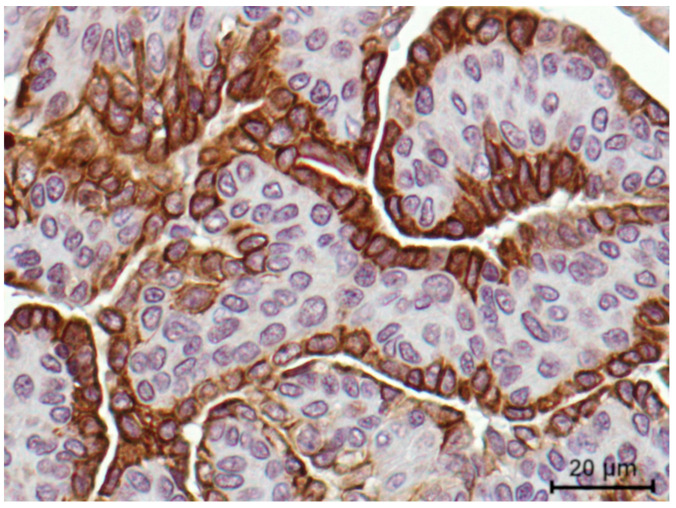
VISTA immunostaining intensity was revealed to be stronger in the epithelial luminal zones of the tumor (3+ score) when compared to the mesenchymal cells, reported in an aggressive feline mammary carcinoma (400× magnification).

**Figure 6 cancers-13-05559-f006:**
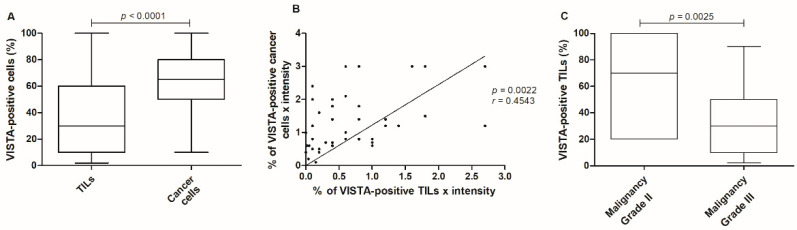
VISTA expression in the feline tumor samples. (**A**) Cancer cells presented a higher VISTA expression when compared to TILs (*** *p <* 0.0001), (**B**) with a correlation obtained between VISTA expression in cancer cells and TILs (** *p =* 0.0022). (**C**) Considering the VISTA-positive TILs population, a higher expression was reported in tumors classified with malignancy grade II (** *p =* 0.0025).

**Figure 7 cancers-13-05559-f007:**
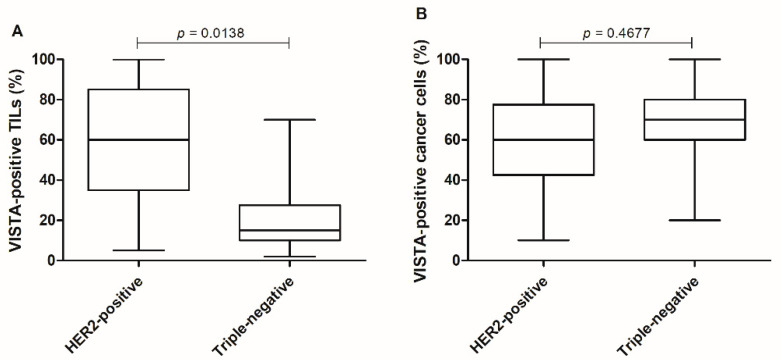
HER2-positive tumors presented higher tissue VISTA expression. (**A**) Categorizing tumor samples by the most aggressive subtypes, HER2-positive and triple-negative tumors, presented differences in the expression of VISTA-positive TILs (* *p =* 0.0138), (**B**) with no differences reported in VISTA-positive cancer cells (*p =* 0.4677).

**Figure 8 cancers-13-05559-f008:**
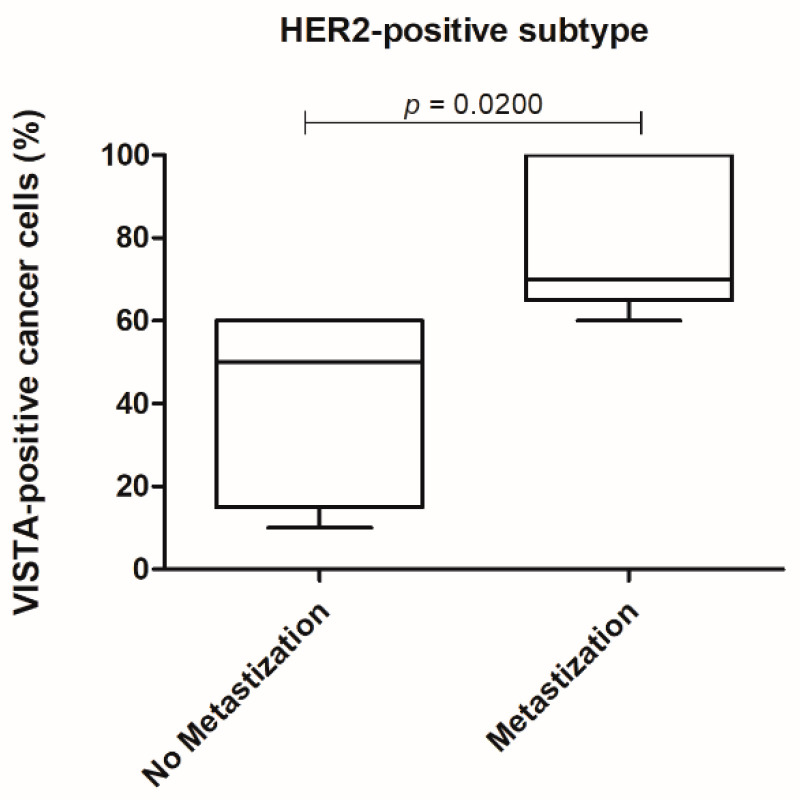
In the HER2-positive tumor subtype a higher VISTA expression in cancer cells was associated with the presence of metastasis (* *p =* 0.0200).

**Table 1 cancers-13-05559-t001:** Clinicopathological features of the 46 queens with mammary carcinomas enrolled in this study (TNM—Tumor, node, metastasis; ER—Estrogen receptor; PR—Progesterone receptor).

Clinicopathological Feature	Number of Animals (%)	Clinicopathological Feature	Number of Animals (%)
**Breed**		**Tumor Size**	
Undifferentiated	33 (71.74%)	<2cm	15 (32.61%)
Siamese	5 (10.87%)	≥2cm	31 (67.39%)
Persian	5 (10.87%)	**Treatment**	
Norwegian Forest	2 (4.35%)	Mastectomy	41 (89.13%)
Russian Blue	1 (2.17%)	Mastectomy + Chemo	3 (6.52%)
**Age**		None	2 (4.35%)
<8years old	3 (6.52%)	**Multiple Tumors**	
≥8 years old	43 (93.48%)	Yes	28 (60.87%)
**Reproductive Status**; 3 unknown		No	18 (39.13%)
Spayed	22 (47.83%)	**Malignancy Grade**	
Not spayed	21 (45.65%)	I	2 (4.35%)
**Lymph node status**; 3 unknown		II	6 (13.04%)
Positive	15 (32.61%)	III	38 (82.61%)
Negative	28 (60.87%)	**Necrosis**	
**Stage (TNM)**		Yes	35 (76.02%)
I	11 (23.91%)	No	11 (23.91%)
II	6 (13.04%)	**Lymphocytic infiltration**; 2 unknown	
III	25 (54.35%)	Yes	29 (63.04%)
IV	4 (8.70%)	No	15 (32.61%)
**Lymphatic Invasion**		**Tumor Ulceration**	
Yes	6 (13.04%)	Yes	7 (15.22%)
No	40 (86.96%)	No	39 (84.78%)
**HER2 Status**		**Ki-67 Index**	
Positive	9 (19.57%)	Low (<14%)	14 (30.43%)
Negative	37 (80.43%)	High (≥14%)	32 (69.57%)
**ER Status**		**PR Status**	
Positive	12 (26.09%)	Positive	24 (52.17%)
Negative	34 (73.91%)	Negative	22 (47.83%)

**Table 2 cancers-13-05559-t002:** Serum VISTA levels (pg/mL) in healthy cats and animals diagnosed with mammary tumors, grouped according to the molecular subtype.

Sample Group	*n*	Median	Mean ± SEM
Control	14	6.01	5.90 ± 0.51
Luminal A	7	9.21	10.31 ± 1.09
Luminal B	16	7.41	10.98 ± 2.32
HER2-positive	10	9.62	12.90 ± 3.73
Triple-negative	13	8.75	14.36 ± 4.16

**Table 3 cancers-13-05559-t003:** Spearman’s correlation between serum VISTA levels and immune inflammation mediators, in cats with HER2-positive, or triple-negative mammary carcinomas.

Serum Immune Checkpoint Proteins and Cytokines Correlation						
**HER2-positive**	**PD-1**	PD-L1	**CTLA-4**	**LAG-3**	**IL-6**	**TNF-α**
**VISTA**	0.0479 *	0.2763	0.1206	0.3088	0.5061	0.0552
**Triple-negative**	**PD-1**	**PD-L1**	**CTLA-4**	**LAG-3**	**IL-6**	**TNF-α**
**VISTA**	0.0005 ***	0.0010 **	0.0083 **	0.0158 *	0.0102 *	0.0083 **

* indicates *p <* 0.05, ** indicates *p <* 0.01, and *** indicates *p <* 0.001.

**Table 4 cancers-13-05559-t004:** Tissue VISTA expression in FFPE samples from FMC, grouped by molecular subtype.

Sample Group	*n*	VISTA-Positive TILs (%)		VISTA-Positive Cancer Cells (%)	
		Median	Mean ± SEM	Median	Mean ± SEM
Luminal A	7	45.00	48.33 ± 10.14	70.00	72.14 ± 8.30
Luminal B	16	30.00	32.94 ± 6.35	60.00	68.75 ± 6.12
HER2-positive	10	60.00	57.22 ± 10.31	60.00	60.00 ± 9.19
Triple-negative	12	15.00	19.75 ± 5.26	70.00	66.92 ± 5.36

## Data Availability

The datasets used and analyzed in the current study are available from the corresponding author in response to reasonable requests.

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
