# Peer review of "VISTA Is a Diagnostic Biomarker and Immunotherapy Target of Aggressive Feline Mammary Carcinoma Subtypes"

_cancers, 2021, doi:10.3390/cancers13215559_

Round 1

Reviewer 1 Report

"VISTA is a diagnostic biomarker and immunotherapy target of aggressive feline mammary carcinoma subtypes" manuscript analyses the use of VISTA (or PD-1H) immunoglobulin as a biomarker for feline mammary carcinoma (similar to the human case).
The paper is well structured, concise, with both methods and results well described and a proper discussion of the results. I would just add the antibodies' ref or cat no. for future reference (particularly, of primary antibodies)!
I find the statistical analysis to have been properly conducted, as well as the histopathological analysis. The different expression of VISTA in the tumor environment is also interesting, and points to an endocrine synergy and immune reactivity. 
Considering the current stand in anatomy-based research and immunotherapy, this work offers important insight for future therapies, as well as what animal models are concerned. Furthermore, this also presents important data for improved treatment of feline mammary carcinomas. This study further suggests VISTA as a negative checkpoint regulator. 

 Author Response

Reviewer 1:

Dear reviewer, thank you so much for your positive remarks on our work. In fact, we are working, not only, to find new therapeutic options for cats, but also to unveil new knowledge about the feline mammary carcinoma, in order to validate the use of cat as an oncology model. Thank you again for your words.

"I would just add the antibodies' ref or cat no. for future reference (particularly, of primary antibodies)!” Thank you for this highlight. Your suggestion was added to the Materials and Methods section, line 191.

Reviewer 2 Report

I really appreciated the aim of the study and I hope to see more about the theme

Author Response

Reviewer 2:

Dear reviewer, thank you for your positive remarks and encouragement to continue this work. In fact, our investigation group is looking for other possible biomarkers in the diseased cats. Our goal is, not only, to find tumor biomarkers that could be useful for diagnosis and as therapeutic targets, but also to prove the similarities between the feline mammary carcinoma and human breast cancer, showing the usefulness of the cat as an oncology model. Thank you once again.

Reviewer 3 Report

In the manuscript entitled " VISTA is a diagnostic biomarker and immunotherapy target of 2 aggressive feline mammary carcinoma subtypes", authors evaluate the expression of the VISTA in feline mammary carcinomas, demonstrating VISTA is a biomarker feline malignant mammary tumors. The paper is rather concise and contains only a few experiments, but, I believe the data are impactful to the field, given the clear need for efficient biomarkers and treatments for feline mammary cancer. 

Author Response

Reviewer 3:

Dear reviewer, thank you so much for your positive remarks and to understand the impact of this type of studies and data. The main goal of our investigation is not only to propose new tumor biomarkers and possible therapeutic targets for cats, but also to prove the value of the cat as a human oncology model, revealing the similarities between both species. As requested, the Results section was revised in order to improve the final quality of the manuscript.

Reviewer 4 Report

In this manuscript entitled "VISTA is a diagnostic biomarker and immunotherapy target of aggressive feline mammary carcinoma subtypes.", the authors investigate the association between VISTA expression and FMC progression or TIL status.

They mainly show that:

  • High serum VISTA levels were higher in feline HER-2 positive and triple-negative tumors.
  • Serum VISTA levels were higher in luminal A tumors that showed metastasis.
  • Serum VISTA levels in cats with triple-negative mammary tumors correlated with immune checkpoint molecules and inflammatory cytokines.
  • The population of VISTA-positive TILs in HER2-positive tumors presented higher than those in triple-negative tumors.
  • The higher population of VISTA-positive cancer cells in HER2-positive tumors was associated with metastasis.

They present many data and claim that VISTA is useful as a biomarker and therapeutic target for feline mammary tumors.

However, most of the serum VISTA levels they assayed in feline mammary tumors were much lower than serum levels of other immune checkpoint molecules and inflammatory cytokines (Fig. 3). Therefore, the reviewer cannot find the superiority of serum VISTA as a biomarker for feline mammary tumors compared to such as PD-1 and CTLA4, and the authors do not mention this. Furthermore, the reviewer feels the consistency of the results of each measurement data is not clear and does not seem to be consistently explained through this manuscript.

Author Response

Reviewer 4:

Thank you for your remarks about our manuscript. Your comments were taking into account in the revision. We hope that after the corrections you consider this manuscript suitable for publication.

“Therefore, the reviewer cannot find the superiority of serum VISTA as a biomarker for feline mammary tumors compared to such as PD-1 and CTLA4, and the authors do not mention this. Furthermore, the reviewer feels the consistency of the results of each measurement data is not clear and does not seem to be consistently explained through this manuscript.”

Dear reviewer, in this work we don’t want to demonstrated that VISTA is superior in comparison to other biomarkers (e.g. PD-1, CTLA-4), we just want to show that VISTA is a suitable tumor biomarker for the diagnosis of feline mammary carcinoma and a promising therapeutic target. Indeed, although VISTA exhibits lower serum concentrations then the ones reported for PD-1 and CTLA-4, the ROC curves of VISTA show higher sensibility to identify the tumor-positive animals, in comparison to PD-1 and PD-L1 ROC curves (doi: 10.3390/cancers12061386). Once this information is important, we added to the manuscript between the lines 460-466.

In addition, we also consider that it is important to increase the number of promising therapeutic target by reporting new and different molecular targets, in order to reduce collateral therapy effects and to surpass the drug-resistance described by using other anti-cancer drugs (e.g. anti-PD-L1 antibodies). Thus, different therapeutic protocols will allow several possibilities and combinations to treatment mammary tumors in a more efficient fashion. This information and 2 new references were added to the manuscript (lines 390 to 395).

Furthermore, Materials and Methods and Results sections were revised in order to improve the final quality of the manuscript.

Round 2

Reviewer 4 Report

The authors have satisfyingly answered my questions. The manuscript is improved and becomes very clear.